# Safety and Potential Complications of Facial Wrinkle Correction with Dermal Fillers: A Systematic Literature Review

**DOI:** 10.3390/medicina61010025

**Published:** 2024-12-27

**Authors:** Audra Janovskiene, Deividas Chomicius, Dominykas Afanasjevas, Zygimantas Petronis, Dainius Razukevicius, Egle Jagelaviciene

**Affiliations:** 1Department of Maxillofacial Surgery, Lithuanian University of Health Sciences, Eiveniu Street 2, 50161 Kaunas, Lithuaniapetronis.zygimantas@gmail.com (Z.P.);; 2Faculty of Odontology, Lithuanian University of Health Sciences, Eiveniu Street 2, 50161 Kaunas, Lithuania; 3Department of Dental and Oral Pathology, Lithuanian University of Health Sciences, Eiveniu Street 2, 50161 Kaunas, Lithuania; egle.jagelaviciene@lsmu.lt

**Keywords:** dermal fillers, soft tissue fillers, complications, adverse events, treatment, management

## Abstract

*Background and Objectives*: The history of facial fillers is very broad, ranging from the use of various materials to modern technologies. Although procedures are considered safe, complications such as skin inflammation, infection, necrosis, or swelling may occur. It is crucial for specialists to be adequately prepared, inform patients how to prepare for corrective procedures, adhere to high safety standards, and continually educate. The goal of this systematic review is to identify complications arising during facial wrinkle correction procedures, as well as to explore safety and potential prevention strategies. *Materials and methods*: The review of the scientific literature followed the PRISMA guidelines. The search was performed in a single scientific database: PubMed. Considering predefined inclusion and exclusion criteria, articles evaluating the safety of dermal fillers used for facial wrinkle correction, complications, and treatment outcomes were selected. The chosen articles were published from 15 February 2019 to 15 February 2024 (last search date: 25 February 2024). The selected articles compared the complications, product safety, and result longevity of various dermal fillers used for facial wrinkle correction. *Results*: In thirty-eight articles, which involved 3967 participants, a total of 8795 complications were reported. The majority of complications occurred after injections into the chin and surrounding area (n = 2852). Others were reported in lips and the surrounding area (n = 1911) and cheeks and the surrounding area (n = 1077). Out of the 8795 complications, 1076 were adverse events (AE), including two severe AE cases: mild skin necrosis (n = 1) and abscess (n = 1). There were no cases of vascular occlusion, visual impairment, or deaths related to the performed procedures. A total of 7719 injection site reactions were classified as mild or temporary, such as swelling (n = 1184), sensitivity (n = 1145), pain (n = 1064), bleeding (n = 969), hardening/stiffness (n = 888), nodules/irregularities (n = 849), and erythema (redness) (n = 785). *Conclusions*: Facial wrinkle correction procedures are generally safe and effective and the results can last from 6 to 24 months, depending on the dermal filler material and its components used. The most common complications after dermal filler injection usually resolve spontaneously, but if they persist, various pharmacological treatment methods can be used according to the condition, and surgical intervention is generally not required.

## 1. Introduction

In 2019, approximately 1.6 million dermal filler injections were performed, representing a 78% increase compared to 2012 [1]. Although filler procedures are considered safe, there are potential risks and complications. According to a study conducted by Akash A. Chandawarkar et al., complications occur at a rate of 1:100 to 1-4:10000, depending on the filler material used [2]. Complications associated with the use of dermal fillers are relatively rare, as evidenced by the steadily increasing number of procedures performed annually in the United States [3]. The most frequently encountered complications include inflammation (16.0%), swelling (14.1%), infection (13.4%), pain (7.9%), and erythema (5.5%), while necrosis accounted for 3.5% of adverse events. Analysis indicates that the proportion of complications such as swelling, pain, and erythema has decreased in the MAUDE (Manufacturer and User Facility Device Experience, USA) database, but there is greater concern over reports of severe complications, such as infection and tissue necrosis [1]. Although such complications are rare, undesirable outcomes, including esthetic (deformities, scarring, etc.) and psychological effects, can be long-lasting [4,5].

Facial dermal fillers are substances injected into the skin to restore lost volume, smooth lines and wrinkles, and enhance facial contours. They can be classified based on their composition, longevity, and application area. Here is a detailed classification [6,7,8,9,10]:
Based on Composition
1.1Hyaluronic Acid (HA) Fillers
Description: HA is a naturally occurring substance in the skin that helps maintain hydration and volume. These fillers are popular due to their natural-looking results and reversibility with hyaluronidase.Longevity: 6 to 18 months.
1.2Calcium Hydroxylapatite (CaHA) Fillers
Description: CaHA is a mineral-like compound found in bones. These fillers provide a more robust fill and stimulate collagen production.Longevity: Up to 18 months.
1.3Poly-L-Lactic Acid (PLLA) Fillers
Description: PLLA is a biodegradable synthetic substance. These fillers work by stimulating collagen production, resulting in gradual and long-lasting volume restoration.Longevity: Up to 2 years.
1.4Polymethylmethacrylate (PMMA) Fillers
Description: PMMA is a biocompatible synthetic substance. These fillers provide permanent volume correction by forming a collagen scaffold around the microspheres.Longevity: Permanent.
1.5Autologous Fat Injections
Description: Fat is harvested from the patient’s body (typically abdomen or thighs) and then purified and injected into the face.Longevity: Permanent with a variable resorption rate.Based on Longevity
2.1Temporary Fillers
Examples: Hyaluronic Acid FillersDescription: Lasts 6 to 18 months, providing temporary volume and wrinkle reduction.
2.2Semi-Permanent Fillers
Examples: Calcium Hydroxylapatite, Poly-L-Lactic AcidDescription: Lasts 1 to 2 years, offering longer-lasting results but not permanent.
2.3Permanent Fillers
Examples: Polymethylmethacrylate, Autologous Fat InjectionsDescription: Provides long-lasting to permanent results, ideal for patients seeking a more enduring solution.Based on Application Area
3.1Lip Fillers
Description: Specifically formulated to add volume and definition to the lips.
3.2Cheek Fillers
Description: Designed to restore volume to the cheeks and midface area.
3.3Nasolabial Fold Fillers
Description: Target the deep lines running from the nose to the mouth.
3.4Under-Eye Fillers
Description: Used to treat tear trough deformities and hollows under the eyes.
3.5Chin and Jawline Fillers
Description: Enhance the chin and define the jawline for a more contoured appearance.
3.6Forehead and Temple Fillers
Description: Restore volume and smooth out wrinkles in the forehead and temple areas.

### Classification of Complications Associated with Facial Dermal Fillers

Facial dermal fillers, while generally safe, can lead to various complications. These complications can be categorized based on their timing (immediate, early, and late), severity, and nature (esthetic, inflammatory, and systemic) [9,11].

Based on Timing
1.1Immediate Complications (Within hours to days)
Pain and Discomfort: Mild to moderate pain at the injection site.Redness and Swelling: Common, typically subsiding within a few days.Bruising: Caused by needle trauma, usually resolves within a week.Allergic Reactions: Rare, can cause localized or systemic reactions.
1.2Early Complications (Within days to weeks)
Infection: Redness, swelling, and tenderness persisting beyond a few days may indicate infection.Nodules and Lumps: Palpable lumps under the skin due to filler material or inflammation.Asymmetry: Uneven results from the filler distribution.
1.3Late Complications (weeks to months)
Granulomas: Chronic inflammatory nodules that can form around the filler material.Migration: Movement of the filler from the original injection site.Persistent Edema: Ongoing swelling in the treated area.Based on Severity
2.1Mild Complications
BruisingTemporary Redness and SwellingMinor Discomfort
2.2Moderate Complications
InfectionPersistent NodulesAsymmetry
2.3Severe Complications
Vascular Compromise: Includes tissue necrosis due to accidental injection into blood vessels, leading to tissue death.Blindness: Rare but serious, resulting from filler entering retinal arteries.Severe Allergic Reactions: Anaphylaxis, requiring immediate medical attention.Based on Nature
3.1Esthetic Complications
Overfilling: Excessive filler resulting in an unnatural appearance.Underfilling: Insufficient filler leading to inadequate correction.Irregularities and Contour Deformities: Uneven distribution of filler.
3.2Inflammatory Complications
Acute Inflammation: Immediate response to the injection.Chronic Inflammation: Long-term response, often leading to granuloma formation.Infection: Bacterial infection requiring antibiotics or drainage.
3.3Systemic Complications
Allergic Reactions: Localized (e.g., itching, redness) or systemic (e.g., anaphylaxis).Vascular Complications: Vascular occlusion leads to tissue necrosis or more severe outcomes like blindness.Autoimmune Responses: Rare, but fillers can trigger autoimmune responses in susceptible individuals.

The aim of this systematic review of the scientific literature is to examine the data presented in scientific sources about facial dermal fillers and investigate the safety and potential complications associated with facial wrinkle correction procedures.

## 2. Materials and Methods

### 2.1. Systematic Review Protocol

This study was conducted in alignment with the guidelines set forth by the Preferred Reporting Items for Systematic Reviews and Meta-Analyses (PRISMA). The PICO methodology was applied to formulate the research question, considering the study’s population, intervention, control, and outcomes, and the problem question was developed (Table 1).

### 2.2. Eligibility Criteria for Articles

#### 2.2.1. Inclusion Criteria

Scientific articles not older than 5 years were selected. Studies are described in full articles in English and Lithuanian. Studies involving people who underwent facial wrinkles correction with dermal fillers, evaluating safety and frequency of complications, and applying complication treatment. Randomized controlled trials and review studies. Studies involving humans.

#### 2.2.2. Exclusion Criteria

Studies with fewer than 30 subjects. Studies on wrinkle correction fillers outside the facial area. Systematic reviews of scientific literature, meta-analyses, case studies, poster presentations, conference presentations, abstracts, expert opinions. Studies involving animals, deceased individuals, or models. Studies describing complications not related to facial dermal filler injections.

### 2.3. Search Strategy

The search for scientific publications suitable for the systematic review of scientific literature was conducted by two researchers, with a third researcher consulted in case of differences or conflicts during the research. The search was carried out in four scientific literature databases: PubMed, Medline, ScienceDirect, and Cochrane. Selected articles were published from 1 February 2019 to 1 February 2024 (the last data search was conducted on 3 March 2024).

The selection of publications was conducted in two stages. In the first stage, scientific articles that did not meet the inclusion criteria were excluded, and titles and abstracts were reviewed to remove publications not relevant to the chosen topic. In the second stage, the full texts of the articles were read, analyzed, and either included in the systemic review or excluded based on the established inclusion and exclusion criteria.

### 2.4. Risk of Bias Assessment in Research

The risk of bias assessment in randomized trials was conducted using the “RoB 2: A Revised Tool for Assessing Risk of Bias in Randomized Trials” questionnaire [12]. This tool is designed to evaluate the level of bias in a publication and its suitability for inclusion in systematic reviews when assessing randomized trials.

The assessment of bias in non-randomized studies of interventions was conducted using the “Risk of Bias in Non-randomized Studies of Interventions (ROBINS-I) tool” questionnaire [13]. This tool is designed to evaluate the level of bias in a publication and its suitability for inclusion in systematic reviews when assessing non-randomized studies.

### 2.5. Criteria for Evaluating the Results of Articles

The following criteria were evaluated in selected scientific publications:Injection site reactions (ISR), a type of adverse event (AE) occurring specifically at the site where facial filler was injected. Common examples include redness, swelling, and pain. These reactions are typically mild and resolve spontaneously.Adverse Event (AE) is a broad term describing any unwanted occurrence, whether in appropriately or inappropriately administered facial filler. Therefore, all ISRs are AEs, but not all AEs are ISRs. For instance, prolonged ISR beyond physiological norms can become an AE. All adverse reactions can range in severity from mild (redness, swelling, etc.), to moderate (asymmetry, nodules, etc.), to severe (allergic reactions, vascular occlusion, blindness, infection, etc.), as well as short-term and long-term. AEs can manifest throughout the body and are not limited to the injection site.Safety of products used in facial wrinkle correction procedures during clinical trials. Safety evaluation during clinical trials is crucial for several reasons: safeguarding consumer welfare, compliance with regulatory requirements, development of improved products, long-term effects, and others.

### 2.6. Data Systematization and Analysis

#### Data Search Results

During the initial stage of searching scientific publications using a selected combination of keywords, 1418 publications were found. Applying selection criteria (publications not older than 5 years, not systematic reviews or meta-analyses, another type of review, and no duplicates), 65 articles were obtained. During this stage, the titles and abstracts of these publications were reviewed. After this stage, publications were excluded due to a mismatch with the topic (n = 16), leaving 49 publications that were selected for full-text analysis.

During the second stage of searching scientific articles, three publications were excluded because their full text was inaccessible, resulting in 46 articles being read and analyzed. Applying exclusion criteria, publications were rejected because the study included fewer than 30 participants (n = 8). A total of 38 publications were included in the systematic review [14,15,16,17,18,19,20,21,22,23,24,25,26,27,28,29,30,31,32,33,34,35,36,37,38,39,40,41,42,43,44,45,46,47,48,49,50,51]. The search process diagram, following PRISMA requirements, is presented in Figure 1.

### 2.7. Risk of Bias Evaluation

Systematic risk of bias assessment utilized the following tools: “RoB 2: a revised tool for assessing risk of bias in randomized trials” for randomized trials, and the “Risk Of Bias In Non-randomized Studies of Interventions (ROBINS-I) tool” for non-randomized clinical studies. The majority (n = 27, 84.38%) of selected randomized trials exhibited low risk of bias, with 15.62% (n = 5) showing moderate risk [32,44,47,49,51]. All non-randomized clinical studies showed moderate risk of bias (n = 6, 100%). Randomized and non-randomized studies with moderate risk of reporting bias did not influence the conducted systematic review. A visual assessment of systematic biases in randomized and non-randomized studies, using the “Risk-of-bias Visualization (robvis)” tool [52], is presented in Figure 2 and Figure 3.

## 3. Results

### 3.1. Characteristics of Studies

This systematic review included 38 studies [14,15,16,17,18,19,20,21,22,23,24,25,26,27,28,29,30,31,32,33,34,35,36,37,38,39,40,41,42,43,44,45,46,47,48,49,50,51]. The majority (n = 32) of selected studies were randomized controlled trials, and six were non-randomized clinical trials [20,21,26,36,37,50]. Most studies had two allocated groups: one control and one experimental, while several studies did not provide complete information [20,21,22,36,37,50].

All studies examined procedures for facial wrinkle correction, detailing the type of filler and technique used, their safety, and reported procedure-related complications. Detailed characteristics of the studies included in the review are presented in Table 2.

### 3.2. Characteristics of the Study Population

In total, 3967 patients were studied, including 243 (6.13%) men and 3621 (91.28%) women; three studies did not provide this data [23,40,51], accounting for 103 participants (2.60%). Authors reported age distribution differently across studies, some providing only means while others varied in age range, thus this criterion was not included in the systemic review.

The most common inclusion criteria in the articles for patient enrollment were the presence of facial wrinkles under study, absence of systemic diseases, no prior corrective procedures, absence of scars or tendency for scarring, and known allergies to materials used in the study (e.g., lidocaine or hyaluronic acid).

The most common exclusion criteria in the articles for patient exclusion were pregnancy, breastfeeding, planning to conceive during the study period, use of certain medications (e.g., anticoagulants), known hypersensitivity, and other factors that could influence study results such as facial inflammation and prior specific facial procedures (e.g., laser or chemical facial procedures). Other authors additionally specified these exclusion criteria: psychiatric disorders or emotional instability [30], sun-damaged facial skin [32,36], smoking [39].

In five studies (n = 5; 13.16%), inclusion or exclusion criteria were not specified [23,25,37,43,44].

### 3.3. Review of Studied Areas, Materials, Sites of Injection, and Instruments

Based on the data from studies included in the systematic review, the majority of facial wrinkle correction procedures utilized materials based on hyaluronic acid (n = 30; 78.95%). Other materials used in studies included polycaprolactone (n = 3; 7.89%), poly-L-lactic acid (n = 2; 5.26%), and others (n = 3; 7.89%).

The most common areas chosen for wrinkle correction in studies were the nasolabial folds (n = 17). Less frequently treated areas included cheeks and surrounding regions (n = 8), lips and perioral wrinkles (n = 7), chin and surrounding area (n = 5), periorbital area (n = 2), inter-brow area (n = 3), mid-face region (n = 3), and lateral orbital area (n = 2). Eleven studies selected more than one facial wrinkle [20,21,24,25,26,31,32,35,36,37,44]. Detailed characteristics of the studies included in the review are presented in Table 2.

The most common site for injecting fillers for facial wrinkle correction was the dermis (n = 16), followed by the subcutaneous layer (n = 14), periosteum (n = 8), and other locations (n = 3). Ten studies did not specify the injection site. Thirteen studies involved multiple injection sites [14,17,18,21,22,24,25,26,27,30,34,35,42].

A 27G needle was most frequently used (n = 15) for injecting the study material into the target area. Other needle sizes used were 30G (n = 7), 32G (n = 4), and 25G (n = 3), and the instrument was unspecified in 12 studies. Several studies employed cannulas of varying sizes (n = 3). Six studies allowed for the use of multiple needle or cannula sizes [14,21,25,34,44,49]. Needle lengths were mostly unspecified (n = 23) in the studies, while 14 studies used ½ inch (12.7 mm) needles, and in 2 studies, needles of lengths 1.0–1.5 inches (25.4–38.1 mm) and 3/16 inch (4.76 mm) were used.

### 3.4. Overview of Complications Reported in Studies

In the studies, injection site reactions (ISR) and adverse events (AE) were recorded. ISRs are interpreted as potential physiological responses and are typically anticipated before corrective procedures, while AEs are complications exceeding those anticipated in studies or prolonged ISRs and all other related phenomena associated with the study.

This systemic review identified 8795 complications of varying degrees, with 7719 ISRs and 1076 AEs. The distribution of Injection Site Responses among selected studies included: edema (n = 1184; 15.34%), sensitivity (n = 1145; 14.83%), pain (n = 1064; 13.78%), hematoma (n = 969; 12.55%), induration (n = 888; 11.50%), nodules/irregularities (n = 849; 11.00%), erythema (n = 785; 10.17%), pruritus (n = 439; 5.69%), changes in color or pigmentation (n = 324; 4.20%), and other (n = 72; 0.93%).

Among the adverse events (AEs) recorded in selected studies, the distribution was as follows: pain (n = 211; 19.61%), hematoma (n = 210; 19.52%), edema (n = 194; 18.03%), erythema (n = 121; 11.25%), nodules/irregularities (n = 54; 5.02%), sensitivity (n = 35; 3.25%), lump (n = 33; 3.07%), induration (n = 27; 2.51%), color changes (n = 25; 2.32%), headache or migraine (n = 22; 2.04%), pruritus (n = 19; 1.77%), and other (n = 125; 11.62%). AEs in the “other” category, comprising 11.62% (n = 125), included: acne (n = 16; 1.49%), nodule (n = 16; 1.49%), bleeding (n = 10; 0.93%), paresthesia (n = 9; 0.84%), cellulitis (n = 7; 0.65%), inflammation (n = 7; 0.65%), speech disorder (n = 6; 0.56%), abscess (n = 6; 0.56%), and other AEs (n = 48; 4.46%) such as petechiae (n = 3; 0.28%), rhinitis (n = 2; 0.19%), hemorrhage (n = 2; 0.19%), cough (n = 1; 0.09%), cyst (n = 1; 0.09%), bronchitis (n = 1; 0.09%), papule (n = 1; 0.09%), toothache (n = 1; 0.09%), etc. Mild necrosis was observed only once (n = 1; 0.09%) among AEs in all selected studies. In one study, one patient required hospitalization due to an AE [14]. There were no recorded deaths, vision impairments, or complete losses of sensitivity in the study area.

In two studies focusing on nasal and peri-nasal corrections, 2755 ISRs and 97 AEs [14,18] were observed, comprising 32.43% (n = 2852) of all complications in the systemic review. Distribution of ISRs: sensitivity—517, swelling—439, pain—428, induration—373, nodules/irregularities—305, hematoma—290, itching—233, erythema—108, and color changes—62. Distribution of AEs: pain—20, lump—10, inflammation—7, swelling—7, abscess—6, acne—6, cellulitis—6, speech disorder—6, and other—29.

For corrections performed on the lips and peri-labial areas, 1911 documented complications (21.73%) were recorded: ISR—1790 and AE—121 [16,17,22,39,40,43]. Distribution of ISRs: hematoma—276, swelling—258, erythema—221, nodules/irregularities—217, induration—217, sensitivity—213, pain—146, color changes—135, and other—107. Distribution of AEs: hematoma—32, pain—27, nodules/irregularities—26, swelling—19, and other—17.

Following mid-facial area corrections (cheeks and peri-cheek areas), 1077 complications (12.25%) were documented: 918 ISRs and 159 AEs [27,32,35,41,42]. Distribution of ISRs: erythema—135, swelling—125, nodules/irregularities—124, hematoma—116, pain—106, sensitivity—106, induration—95, color changes—66, and itching—45. Distribution of AEs: pain—60, swelling—33, erythema—22, hematoma—14, headache—9, and other—21.

In the nasolabial fold area, 744 complications (8.46%) occurred, with 600 ISRs and 144 AEs [15,28,30,33,38,45,47,48,49]. Distribution of ISRs: pain—160, swelling—127, sensitivity—83, erythema—81, itching—64, hematoma—52, papule—26, and color changes—7. Distribution of AEs: swelling—63, color changes—18, erythema—15, pain—15, hematoma—9, sensitivity—7, and other—17.

In other facial areas, following wrinkle correction procedures, 2211 (25.14%) complications were documented: 317 (3.60%) in the periocular region [34,50,51], 29 (0.33%) in the lateral orbital area [23], and 1865 (21.21%) in studies where facial fillers were used in multiple facial areas simultaneously.

### 3.5. An Overview of Treatment Methods for Complications Reported in Studies

In most selected studies, information on treatment was predominantly spontaneous (n = 26; 60.47%), without any intervention by researchers, involving resolution or healing of injection site reactions or adverse reactions. Treatment of complications was not mentioned in nine studies (n = 9; 20.93%). Among the reported treatments requiring researcher intervention, the most common was hyaluronidase treatment (n = 3; 6.98%), used for persistent inflammation in the study area, unresolved nodules, and cysts [14,18,24]. Additionally, antibiotic therapy was applied (n = 2; 4.65%), systemic steroids (n = 1; 2.33%), non-steroidal anti-inflammatory drugs (n = 1; 2.33%), and mometasone furoate glucocorticoid (n = 1; 2.33%) [14,24,34,42]. In one study by Kenneth Beer and others, a patient was hospitalized, and an abscess was drained [14].

### 3.6. Overview of Safety Findings from Studies

According to data presented in the conclusions of thirty-eight reviewed studies, the materials used in the studies were mostly evaluated as safe or with minimal safety concerns (n = 29; 76.32%). The effectiveness of the materials used in correcting wrinkles in the facial area under study was mentioned in twenty-two conclusions (n = 22; 57.89%), and they were well tolerated in four conclusions (n = 4; 10.53%).

It was noted in the conclusions that the results of the materials used in the studies last at least 6 months [26,36] but can last for up to 24 months [20,42]. Persistence at 12 months was mentioned in five studies (n = 5; 13.16%) [14,24,27,34,49]. However, the most common procedure’s long-term results were not specified (n = 23; 60.53%).

In seven (n = 7; 18.42%) studies’ conclusions, it is emphasized that additional research or long-term monitoring of the obtained results is necessary [15,16,21,33,36,50,51]. In one study (n = 1; 2.63%), it is mentioned that the use of stereophotogrammetry in further research would be beneficial for more accurately evaluating the results.

## 4. Discussion

This systematic review analyzed 38 scientific publications, including 32 randomized trials and 6 non-randomized clinical trial studies [14,15,16,17,18,19,20,21,22,23,24,25,26,27,28,29,30,31,32,33,34,35,36,37,38,39,40,41,42,43,44,45,46,47,48,49,50,51]. The main objective of this work was to review the data presented in scientific sources about facial dermal fillers and to investigate the safety of facial wrinkle correction and potential complications associated with the procedure.

The majority of dermal fillers studied in the publications (n = 35; 92.11%) were temporary (results typically lasting from 6 to 18 months), two (5.26%) were semi-permanent (results typically lasting up to 24 months and longer) [36,49], and one (2.63%) was permanent [51]. Based on studies conducted by other authors, it can be concluded that as facial tissues continuously change, safer materials are those that can be biologically degraded more quickly by the body over time, resulting in fewer long-term complications [53]. Additionally, many authors consider safer those materials that can be broken down in case of unforeseen complications, such as hyaluronic acid filler with hyaluronidase, thereby avoiding surgical intervention which can leave scars, cause psychological consequences, or lead to other complications associated with such interventions.

Corrective procedures involving facial dermal fillers can lead to adverse events. While most consequences can be mild, such as ISRs (e.g., erythema, swelling, pain) and disappear over time, rare (e.g., vision impairment or tissue necrosis) and severe complications cannot be ignored as their effects can be irreversible (e.g., blindness, scarring) or life-threatening. In this systematic review, there was no significant number of severe complications, such as vascular occlusion, tissue necrosis, vision impairment, and infection, which can significantly impact quality of life and health, as described in the literature and other studies [53,54]. Specialists must be aware of all these risks and properly inform patients about potential risks, as timely detection and appropriate treatment can prevent severe and irreversible consequences.

In this systematic review, the most common negative tissue reactions at the injection site resolved spontaneously, and some adverse events were treated with s, NSAIDs, hyaluronidase, etc. Only one patient (0.003%) required treating due to an abscess that needed special treatment, including intravenous antibiotics and abscess drainage. Thus, it can be concluded that a specialist must have appropriate education, be able to provide emergency care, and know where and when to refer patients to other specialists if severe complications develop.

Common injection site reactions included edema, sensitivity, pain, hematoma, induration/stiffness, nodules/irregularities, and erythema. These accounted for 89.17% of the complications.

Among the adverse events, the most common were pain, hematoma, edema, erythema, nodules/irregularities, sensitivity, and lumps. Together, these accounted for 79.75% of these complications.

The included studies did not establish a connection between the use of cannulas and a reduced risk of complications compared to the use of needles due to the small sample size. Therefore, more studies are needed to evaluate this. According to other researchers, the use of cannulas for filler injection may reduce the incidence of bruising. These authors believe that the sharp tip of the needle can cause more tissue trauma [55]. Although both needles and cannulas can be used for facial corrective procedures, cannulas may be safer and reduce the risk of intravascular complications [56].

The methods for treating complications were varied, but the most frequently occurring complications spontaneously resolved (60.47%) without investigator intervention. In both the studies reviewed in this systemic review and the results of other researchers’ studies, hyaluronidase was often chosen as the first-line treatment for complications related to hyaluronic acid fillers. Concurrently, medical treatment, depending on the condition, included antibiotics, systemic steroids, NSAIDs, and glucocorticoids [57,58].

In most studies (76.32%) [14,15,16,18,19,21,22,23,24,25,26,28,30,32,34,35,36,37,38,39,41,42,43,44,45,46,47,48,49], it was concluded that the materials used were safe or had minimal safety-related risks. These safety conclusions were largely consistent with those of other researchers [53,55].

The longevity of the results of facial wrinkle correction varied, with some studies indicating that the effects lasted from several months to 24 months, which was associated with the type of material used and its quantity.

## 5. Conclusions

The analysis of selected studies on facial wrinkle correction reveals that hyaluronic acid-based materials are the most commonly used. This preference is likely due to the ability to manage varying degrees of complications with hyaluronidase effectively. The nasolabial folds emerged as the most frequently treated anatomical region in these procedures, as highlighted in the systematic review.

While mild, short-term complications were commonly expected and typically resolved on their own, there were instances of moderate to severe complications requiring targeted treatment. Overall, the findings suggest that facial correction procedures are generally safe. However, the outcomes can be significantly influenced by several factors, including the qualifications of the practitioners, the properties of the materials used, variations in techniques and methodologies, and the level of cooperation among patients.

Further research is needed to investigate the long-term safety, effectiveness, and outcomes of these procedures across diverse populations, as well as to explore advanced techniques and materials that may reduce complication rates.

## Figures and Tables

**Figure 1 medicina-61-00025-f001:**
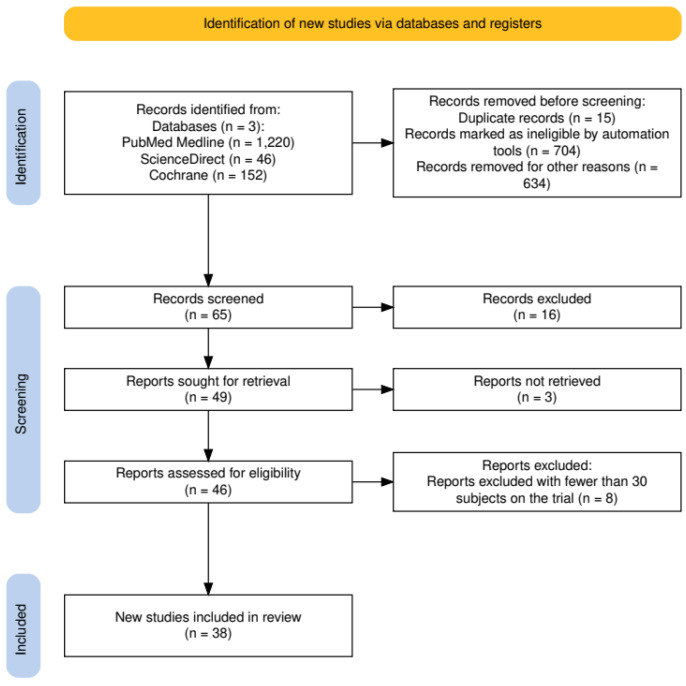
Systematic review article search process diagram.

**Figure 2 medicina-61-00025-f002:**
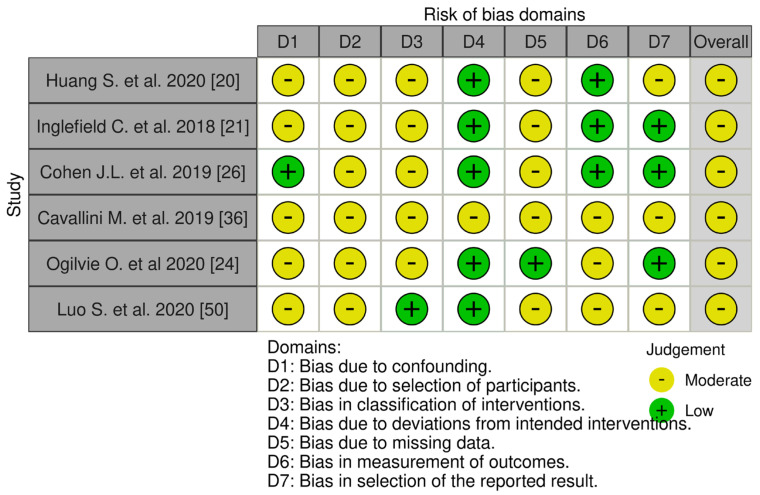
Systematic risk assessment of non−randomized clinical trials [20,21,24,26,36,50].

**Figure 3 medicina-61-00025-f003:**
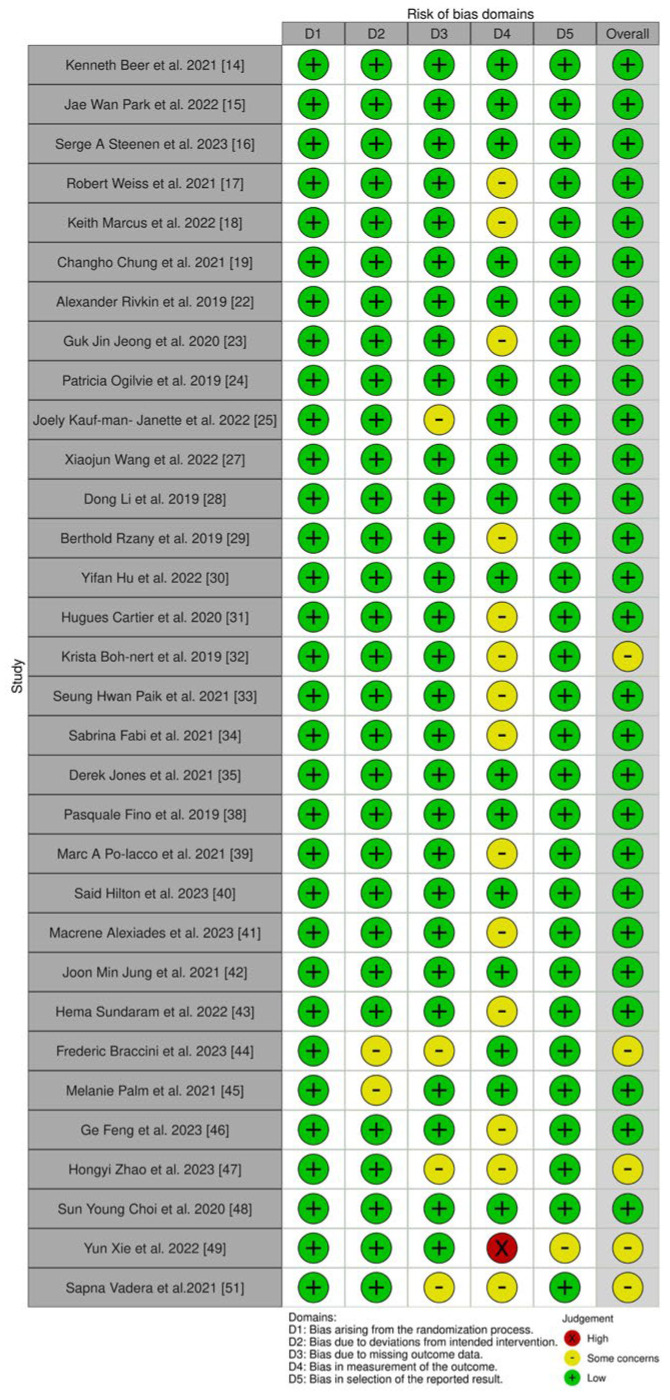
Assessment of Systematic Risk in Randomized Controlled Trials, refs. [14,15,16,17,18,19,22,23,24,25,27,28,29,30,31,32,33,34,35,38,39,40,41,42,43,44,45,46,47,48,49,51].

**Table 1 medicina-61-00025-t001:** Description of the problem question formulation using the PICO analysis method.

PICO Method	Description of the Components of Question Formulation
P—population	Patients undergoing facial rejuvenation procedures
I—intervention	Facial dermal filler injection
C—control	Patients not undergoing facial dermal filler injection or receiving a control substance injection
O—outcome	Complications associated with facial dermal filler procedures, their types, and severity
Research question	Are facial wrinkle correction procedures with fillers safe?

**Table 2 medicina-61-00025-t002:** Summary overview of results from studies included in the systematic review.

Publication	Material	Study Type	Number of Subjects	Subjects/Control	Gender of Subjects	Research Area	Where Dermal Filler Injected and Instrument Used	Complications	Treatment	Conclusions
Kenneth Beer et al. 2021 [14]	HA	RCT	192	144/48	M—22 F—170	Chin (pogonion, menton, prejowl sulci)	Supraperiosteal and/or subcutaneous injections, 27 G 1/2-inch needle and cannulas	AE: 6—abscess, 1—gum pain, 6—acne, 7—cellulite, 7—inflammation, 1—hypoesthesia, 3—lumps/bumps, 3— erythema, 3—pain, 0—deaths and loss of sensitivity. ISR (HA): 277—tenderness to touch, 259—hardness, 226—swelling, 215—pain, 203—lumps/bumps, 189—hemorrhage, 176—itching	AE after initial/improved treatment resolved without sequelae. Injection site inflammation (n =1) and injection site cellulitis. Treatment consisted of AB, anti-inflammatory drugs and hyaluronidase. Injection site erythema (n = 1) lasted 264 days, acne (n = 1) lasted 134 days, spontaneous healing.	The research material is safe and effective, the results remained for 1 year. Less injection site response was observed with the cannula.
Jae Wan Park et al. 2022 [15]	PCL, 2 products	RCT	59	59/59	M—3 F—56	Nasolabial fold	Retrograde injection technique—not specified how deep, 27G needle	AE: test product: 43 reactions to filler (43—swelling, 4—pain, 1—erythema); control: 18 reactions to filler (18—swelling, 3—pain). AE disappeared within 30 min after the procedure.	Not available	The newly developed PCL-based filler SF-01 and the existing Ellansé-M were effective and safe in the correction of NLF. SF-01 is non-inferior to Ellansé-M in correcting moderate to severe NLF. SF-01 is effective. More research is needed.
Serge A Steenen et al. 2023 [16]	HA, 4 produtcs	RCT	143	35/39/37/3 2	M—0 F—143	Lips	-	No serious AEs were recorded. Data on injection site reactions to the filler are not provided.	Not available	A statistically significant but clinically equivocal advantage was found in terms of survival time, patient satisfaction, and quality of life. No product was superior in all categories. Safety indicators were the same. Useful for future research to use stereophotogrammetry.
Robert Weiss et al. 2021 [17]	HA	RCT	273	184/89	M—9 F—261	Lips and above lips area	Submucosal layer of the lips, or in the mid-dermis to subcutaneous layer of upper perioral rhytids, needles (size not specified)	The most reported (occurring in ≥5% of subjects) were injection site lump (test: 10%; control: 11%), bruising (test: 8%; control: 10%) and nodule (test: 5%; control: 7%). All the most common study-related AEs were mild, except for one case of moderate injection site bruising. The reported frequency of late AEs (occurring ≥21 days) was similar in all groups (test: 5%; control: 6%). All late-onset events were mild (test) or mild to moderate (control) and all resolved or were rated as stable.	Most resolved spontaneously	Study material was non-inferior to control at week 8, well tolerated and effective, persisting in 60% of subjects at week 48 after the last injection. Efficacy of the study material was supported by significant esthetic improvement and subject satisfaction.
Keith Marcus et al. 2022 [18]	HA	RCT	140	107/33	M—15 F—125	Chin and surrounding area	In the pogonion mainly supraperiosteally, surrounding areas mainly subcutaneously, 27G 1/2-inch needle	AE: 8—pain, 3—bruises, 3—swelling, 2—erythema, 2—hemorrhage, 2—nodule. AE lasted on avg. 4 days. There were no serious study-related AEs during the study. The most common injection-related symptoms among subjects were tenderness (90%), pain (74%), and swelling (74%), which usually disappeared within 14 days.	One nodule resolved with hyaluronidase (resolved after 74 days) and one nodule resolved spontaneously after 112 days.	Hyaluronic acid filler was safe and effective in correcting the chin and surrounding areas. Esthetic improvement was demonstrated by both treating investigators and subjects throughout the study. Subjects’ satisfaction rates remained high up to 48 weeks post-last injections.
Changho Chung et al. 2021 [19]	HA	RCT	91	91/91	M—10 F—81	Nasolabial fold	Linear injection technique—not specified how deep, instrument not specified	Local reactions were predictable: redness and swelling. Local reactions usually disappeared within 2 weeks, without any treatment and there were no reports of severe local reactions. No other serious systemic AEs were reported. No significant difference in pain was observed between the groups.	Not available	This study proved that the filler is an effective and safe material for facial rejuvenation. Safety did not differ from the control filler. Adjusted for moderate and severe NLFs, test filler scores were higher than controls.
She-Hung Huang et al. 2020 [20]	HA	NRT	100	-	M—0 F—100	Cheek—100, Nasolabial fold—88, Temporal area—25, Nose—43 and Chin—84	-	Tenderness in the area of the procedure was reported by all subjects and usually subsided after 3 days, but in a few cases, symptoms resolved after 14 days. A total of 260 AEs were recorded in 64 (64%) subjects, among 5 of which were severe. Of the 29 AEs in 16 subjects, none were severe. AE: 13—pain, 7—bruises. All AEs were mild (79%) and moderate (21%).	On average, AE lasted 16 days, all resolved by the time of the return visit.	The results of the study show that the test material and the control material are effective in the treatment of the whole face. This is evidenced by long-term esthetic improvement and high subject satisfaction at 24 months after a single repeat procedure. The data can be used to help develop individualized treatment plans for patients.
Christopher Inglefield et al. 2018 [21]	RPC	NRT	30	-	M—1 F—29	Facial contouring, i.e., Nasolabial fold	Intradermally or subdermally using a thin needle or cannula (27–31G)	80% (24/30) of participants experienced at least one injection site reaction after the initial injection, and 40% (6/15) who presented for a repeat procedure—experienced at least one injection site reaction. All but one of the IVRs were mild to moderate in severity. A severe case of hemorrhage reported in week 1. The total amount of AE was 40: 14—hemorrhages, 31—redness, 14—swelling, 21—sensitivity, 3—itching, 13—hardening.	Injection site reactions resolved by the 1st follow-up visit, with only 31% of participants experiencing injection site reactions 1 week prior to any procedure. Only one injection site reaction was reported in week 4. It was a mild case of induration that disappeared before the last visit. No injection site reactions were reported by the investigator at the last visit (week 12).	This study shows that RPC is safe to use for facial contouring and has a good indication of long-term safety in the absence of signs of hypersensitivity. Performance results are promising for biological skin regeneration using RPC. Extra research is required to confirm long-term results and its use in other areas of the face, not just the NLF.
Alexander Rivkin et al. 2019 [22]	HA	RCT	164	-	M—0 F—164	Lips and surrounding area	Mid to deep dermis, 30G 1/2-inch needle	ISR (n = 120): 112—swelling, 108—tenderness, 102—hardening, 102—lumps/ bumps, 98—bruising, 92—pain, 90—redness, 41—discoloration, 33—itching. The majority of IVAs were mild or moderate (n = 724), severe—170. AE: nodules/bumps (32%), hemorrhages (17.9%), and pain (12.5%). After repeated procedure (17/124, 13.7%): lumps/bumps (7.3%), hardening (4%).	The majority (71.6%) of ISRs reported after re-procedure resolved within 2 weeks. Most study-related adverse events occurred within 1 day after the repeat procedure, were mild to moderate, did not require treatment, and resolved within 60 days, without consequences. One subject (0.8%) reported a treatment-related AE after retreatment, which continued at the end of the study, without requiring treatment.	Repeat treatments with hyaluronic acid after 1 year were safe and effective for correction of the lips and perioral area, and a lower amount of product was required to achieve similar efficacy to the initial treatment.
Guk Jin Jeong et al. 2020 [23]	PCL, 2 products	RCT	30	30/30	-	Lateral area of the eye	Linear injection technique—not specified how deep, 33G needle	ISRs were transient edema (50.00%), pain (23.33%), pruritus (13.33%) and erythema (10.00%). Adverse events occurred at a similar frequency in both groups. No serious systemic AEs reported.	Not available	New PCL-Based Dermal Filler DLMR01 shows adequate efficacy and safety, expands options for clinicians and patients with lateral ocular wrinkles.
Patricia Ogilvie et al. 2019 [24]	HA	RCT	120	90/30	M—0 F—120	Chin and lower jaw contouring	Above the periosteum and/or hypodermis, 27G 13 mm needle	ISR: 114—hardening, 114—tenderness, 109—swelling, 109—pain, 108—redness, 102—lumps/bumps, 101—bruising, 62—discoloration, 57—itching. AE: 10—lump, 9—pain, 4—swelling, 6—speech disorder, 3—induration, 2—headache, 2—nodule, 1—hematoma, 1—cyst, 1—color change, 1—erythema, 1 -itching, 4—other.	Most ISRs disappeared within 1 week after the procedure. Ten subjects had treatment-related AEs at the injection site -started more than 60 days ago five subjects had delayed cyst, nodule(s), or nodule. In two subjects, these events resolved within 2–8 weeks, and in the other two subjects, they persisted for 12 months. One cyst was moderate and resolved within 13 days of treatment with hyaluronidase and topical mometasone furoate; the rest of the events were mild.	Research HA VYC-25L injectable gel has an acceptable degree of safety and is effective for up to 12 months in restoring and increasing facial volume.
Joely Kaufman- Janette et al. 2022 [25]	HA 4 products	RCT	30	30/30	M—0 F—30	Nasolabial fold, Lips and Surrounding area	From deep dermis to superficial subcutaneous tissue, 30G 1/2-inch and 27G 1/2-inch needle	AE NLF: 7—induration, 1—bump, 2—itching, 1—paresthesia. AE around the lips: 4—hardening, 2—pain, 1—bump, 2—needle puncture mark.	Most ISRs disappeared in 14 days or less. Most AEs resolve spontaneously within several days or weeks.	HA fillers with mepivacaine or lidocaine are equally effective in reducing pain and are safe for facial correction wrinkles and folds.
Joel L Cohen et al. 2019 [26]	BA ir HA	NRT	154	80/74	M—12 F—142	Glabella, Forehead, Lateral area of the eye, Nasolabial fold, Lips and	Mid- to deep dermis (unspecified), instrument not specified	BA AE: 2—pain, 1—bronchitis, 1—hematoma, 1—muscle paresis, 2—ptosis, 8—facial paresis, 6—headache and migraine, 2—swelling, 6—other. HA AE: 3—acne, 19—bruises, 2—bump, 4—herpes, 7—sensitivity, 5—other.	Treatment-related AEs were mild in intensity and also resolved during the study.	The efficacy and safety of BA and HA in subjects with moderate to severe facial wrinkles and folds was similar regardless of procedure, and facial esthetic improvement was maintained at least 6 months.
Xiaojun Wang et al. 2022 [27]	HA	RCT	147	110/37	M—12 F—135	Middle face area	Hypoderma and supraepiderma, instrument not specified	Four subjects (2.8%) experienced AE, the most common were: 3—pain, 2—swelling.	All AEs resolved without intervention.	Midface treatments using the investigational HA material were effective, well tolerated, and resulted in high levels of subject satisfaction up to 12 months.
Dong Li et al. 2019 [28]	HA	RCT	100	100/100	M—2 F—98	Nasolabial fold	Deep layer or mid- of dermis and/or surface layer of the subcutis, instrument not specified	Two subjects (2%) experienced an AE related to the study product and/or the injection procedure: transient bruising after the procedure. Both AEs spontaneously disappeared within a week.	Average duration of symptoms was 2–4 days, the symptoms did not worsen after the corrective procedure. No subject continued to have AEs 14 days after the corrective procedure, and no differences were found between study products.	Control materials safety has previously been demonstrated and is shown in this study for the investigational substance Restylane Lyft. The study showed that Restylane Lyft is effective and well tolerated for moderate to severe correction NLF.
Berthold Rzany et al. 2019 [29]	HA, 3 products	RCT	90	30/30/30	M—15 F—75	Nasolabial fold	-	ISR: Bruising, redness, induration, pain/tenderness, lumps/bumps and swelling. After one month after the procedure, only a few subjects experienced redness, induration, a lump or swelling.	Not available	This study demonstrated multiple advantages/benefits of three new HR fillers in dynamic wrinkle correction.
Yifan Hu et al. 2022 [30]	HA	RCT	96	49/47	M—1 F—95	Nasolabial fold	Mid dermis and hypodermis, 27G 1/2-inch needle	AE in group A (Restylane): 10—swelling, 7—redness, 3—sensitivity, 1—hemorrhage. AE in Group B (Yishumei): 12—swelling, 7—redness, 4—sensitivity, 2—bleeding, 1—itching.	All AEs resolved spontaneously in all participants within 7 days without treatment.	Group A (Restylane) and Group B (Yishumei) have a good therapeutic effect in correcting NLF, but Restylane had a longer effect that could last 24 weeks, while Yishumei lasted about 12 weeks. Safety assessments have shown that both materials are safe.
Hugues Cartier et al. 2020 [31]	BA ir HA	RCT	60	29/31	M—1 F—59	Glabella, Cheeks, Nasolabial fold	Botulin—intramuscularly, HA—not specified. Instrument not specified	HA ISR (monotherapy): One subject had bilateral hematomas. AE: most often there were: 22—bruise. In the second combined study, AEs (13 subjects (23%) experienced 18 AEs), all related to HA, mostly there were: 14—hemorrhage. All AEs were mild (n = 15) or moderate (n = 3) and none were difficult.	HA ISR (monotherapy) disappeared after 15 days. In the first combination study, most AEs resolved within 2 weeks, and none were severe. In the second combination study, most AEs resolved within one week.	In this study, combined treatment with BA and HA resulted in more beneficial esthetic results compared to monotherapy with BA or HA. Monotherapy with BA may be more beneficial as a first step than HA with a filler. So much both combination therapy and monotherapy were well tolerated.
Krista Bohnert et al. 2019 [32]	PLLA and SA	RCT	40	20/20	M—0 F—40	Cheeks, Middle face area	Dermal, 25G 1.0-1.5-inch needle	No procedure-related AEs were observed. Swelling occurred after the first PLLA injection 3 to the subjects.	The temporary swelling, rated as mild by the subjects, resolved within a week without sequelae.	The results suggest that retreatment of PLLA does not only improve contour deficits. PLLA safety was similar to that described in the literature.
Seung Hwan Paik et al. 2021 [33]	HA, 2 products	RCT	91	91/91	M—8 F—83	Nasolabial fold	Linear injection technique—not specified as deep, 27G needle	AE was mostly: 15—pain (50%), 5—swelling (16.7%), 4—bruise (13.3%), 2—headache (6.7%), 1—redness (3.3%), 1—itching (3.3%), 1—cough (3.3%).	Both fillers were well tolerated, and AEs were mild and lasted less than 2 weeks. Most of the side effects were expected, of mild intensity and disappeared without consequences.	The test material (BM-PHA) was not inferior to the control (Restylane Perlane). Further studies are needed to confirm the long-term safety of BM-PHA.
Sabrina Fabi et al. 2021 [34]	HA	RCT	135	103/32	M—11 F—124	Area under eye	Above the periosteum or below the muscle layer, 32G 1/2 inch or 27G 1/2-inch needle	ISR: 63—tenderness, 51—hematoma, 55—swelling, 49—lumps/bumps, 46—induration. In the study group, the most frequent AEs (occurred (≥2% of participants) included bruising (3.8%) and swelling/edema (2.9%).	Most ISRs disappeared within 1 week. AEs resolved without sequelae. Three participants (2.9%) experienced three AEs (>30 days post-procedure), swelling or edema; two resolved in ≤4 days, one lasted 45 days and resolved in AB.	The study material VYC-15L was safe and effective. The results lasted for 1 year. Similar efficacy was achieved regardless of the injection instrument (needle or cannula), and safety favored VYC-15L for injections.
Derek Jones et al. 2021 [35]	HA, 2 products	RCT	210	142/68	M—23 F—187	Cheeks and Middle face area	Above the periosteum and hypodermis, 27G needle and cannula	In the study material group AE there were: 16—pain, 5—hematoma, 6—swelling, 6—redness, 2—bleeding. In the control group, AE was: 36—pain, 1—hematoma, 15—swelling, 11—redness, 4—bleeding.	In the control group, there was one severe injection site swelling that resolved without intervention lasting 3 days. The median duration of AE in group A was 3 days in both the test and control groups.	After 12 weeks after the last injection, the test substance was tolerated. esthetic improvement and satisfaction were high and continued up to 48 weeks. No safety issues found. Cannula injections reduce bruising compared to needle injections, but more research is needed.
Maurizio Cavallini et al. 2019 [36]	HA	NRT	40	-	M—0 F—40	Cheeks, area around lips, glabella, forehead area (as needed)	Dermal, 32G 13mm needle	Transient post-procedure erythema (redness) or slight edema (swelling) was observed in three patients and lasted no longer than 12 h. Pain during treatment generally considered tolerable by patients.	Not available	The test substance (VYC-12) improves skin quality as measured by an objective tool (DACS). This study demonstrated the efficacy and safety of VYC-12 in improving facial skin quality and texture up to 6 months.
Patricia Ogilvie et al. 2020 [37]	HA	NRT	128	-	M—15 F—113	Cheeks, forehead and, optionally, neck	Dermis, instrument not specified	ISR: redness (6.1%) and bruising (6.1%). AE was: 12—bump, 4—bleeding, 2—hematoma, 1—erythema, 1—nodule. ISR after the first procedure: 127—redness, 121—swelling, 118—sensitivity, 114—hematoma, 114—stiffness, 112—lumps/bumps, 107—pain, 39—itching, 38—color changes. ISR after repeated procedure: 53—redness, 44—swelling, 45—sensitivity, 46—hemorrhage, 43—stiffness, 42—lumps/bumps, 42—pain, 13—itching, 16—color changes.	Most ISRs disappeared within 1 week.	Subjects using the study material (VYC-12) expressed satisfaction with the reduction in fine lines and other skin quality features, also after repeated treatments. Most subjects experienced minimal discomfort after the procedures. Acceptable safety and tolerability profile.
Pasquale Fino et al. 2019 [38]	HA, 2 Products	RCT	65	65/65	M—3 F—62	Nasolabial fold	Dermis, instrument not specified	With an additional injection after 1 month. (BB 8/11, 72.7% vs. ISD 8/10, 80.0%). During the first injection, 50 subjects in each group reported pain as mild or moderate (76.9%). Redness after the first injections (BB 46/65, 70.8% vs. ISD 47/65, 72.3%). Hemorrhage was more frequent after ISD than BB at 14-day follow-up (BB 16/64, 24.0% vs. ISD 26/64, 40.6%).	Not available	ISD achieves a long-term result (more than 9 months) in correcting moderate to severe wrinkles, similar to BB. Both products had a high degree of safety and a high level of acceptance by participants and researchers’ degree of satisfaction.
Marc A Polacco et al. 2021 [39]	HA, 2 products	RCT	48	48/48	M—1 F—47	Wrinkles around the lips	Dermal, 32G needle	AE (SP-HAL): 2—color changes, 1—nodule. AE (CPM-HA): 1—rash, 1—acne.	All AEs resolved spontaneously.	While SP-HAL and CPM-HA effectively reduces wrinkles around the lips and has a similar safety profile, SP-HAL’s result was sustained longer.
Said Hilton et al. 2023 [40]	HA, 2 Products (HA_RK_ or HA_JUS_)	RCT	40	20/20	-	Lips	Hypodermis, 30G 1/2-inch needle	AE was pain, papules and swelling. AE (HA_RK_): 19—swelling, 7—redness, 18—hemorrhage, 14—pain/sensitivity, 1—itching. AE (HA_JUS_): 20—swelling, 15—redness, 17—hemorrhage, 17—pain/sensitivity, 5—itching.	Not available	The intensity of early swelling and other ISRs was lower with HA_RK_ than with HA_JUS_, the difference may be related to differences in filler properties. However, esthetic improvement, subject satisfaction and persistence of AE after 14 days were similar.
Macrene Alexiades et al. 2023 [41]	HA, 2 products	RCT	202	131/71	M—28 F—174	Cheeks and surrounding area	Dermis, 32G 1/2 inch or 32G 3/16 inch needle	AEs were: 9—headache (4.7%) and 6—acne (3.1%). Other AEs occurred in <2% of participants. Six participants’ (3.0%) AE were 3—itching (1.5%), 2—redness (1.0%). One participant developed two mild papules at the injection site at day 117 after injection. ISR (study group, n = 135): 94—redness, 87—swelling, 85 —lumps/bumps, 81—hematoma, 76—pain, 76—sensitivity, 67—stiffness, 47—color changes, 29—itching, 12—others. ISR (control, n = 64): 41—redness, 35—swelling, 39 —lumps/bumps, 35—bruising, 30—pain, 30—tenderness, 28—stiffness, 19—discoloration, 16—itching, 13—other.	One participant developed severe bruising at the injection site that resolved without intervention. Papules formed after 117 days disappeared by the end of the study. Most ISRs were mild to moderate and resolved within 7 days.	Investigational VYC-12L is an effective, well-tolerated, long-term improvement in cheek skin smoothness, fine lines and hydration. The safety of buccal injection of VYC-12L is similar to previous studies of VYC-12L and other HA fillers.
Joon Min Jung et al. 2021 [42]	HA, 2 products	RCT	69	69/69	M—15 F—54	Cheeks and surrounding area	Above the periosteum and hypodermis, 27G needle	AE (research materials): 6—swelling, 3—lump, 3—pain, 1—hematoma, 2—redness, 1—itching. AE (control): 4—swelling, 4—bump, 2—pain, 2—hemorrhage, 1—redness, 1—itching. All AEs were mild to moderate in severity.	At the end of the 24-week study period, subject 1 experienced delayed recurrent swelling of both sides of the face. The symptom first appeared 28 weeks after the initial injection. She was treated with systemic steroids and her symptoms completely disappeared within 12 weeks.	This study demonstrated similar durability and safety of the study SHAPE-NVL and control material VYC-20L over a 2-year period. Both types of HR fillers have a long-term effect of up to 2 years, with minimal safety issues.
Hema Sundaram et al. 2022 [43]	HA	RCT	202	150/52	M—4 F—198	Wrinkles around the lips	Dermis, 30G 1/2-inch needle	ISR: 131—redness, 54—pain, 105—sensitivity, 115—stiffness, 146—swelling, 115—lumps/bumps, 154—hemorrhage, 31—itching, 94—color changes. There were no hard AEs.	All AEs were mild to moderate, resolved, and none were considered clinically significant.	The investigational material RHA is effective and safe for the correction of wrinkles around the lips. The results of this study are consistent with previous AI studies that have demonstrated the safety, efficacy, and long-term effects of other RHA fillers on esthetic outcomes. for facial procedures.
Frederic Braccini et al. 2023 [44]	HA, 2 products	RCT	98	98/98	M—6 F—92	Middle face area, lower edge of the lower jaw, chin, temporal area	Above the periosteum, 27G 1/2-inch 13mm needle or 25G, 55mm cannula	AE (research material): 11—redness, 2—hematoma, 3—hematoma, 1—lumps/bumps, 23—pain. AE (control): 13—redness, 2—hematoma, 7—hematoma, 1—edema, 2—discoloration, 1—lumps/bumps, 1—necrosis (mild), 25—pain.	All AEs were expected and routinely observed with HA-based fillers. Most of these phenomena have disappeared at day 21 and throughout the study, no severe AEs were reported with either filler.	This study confirmed the efficacy and safety of the investigational HA filler compared to a control with efficacy lasts at least 18 months. The assessment methods used in the study confirmed that it was non-inferior to the CE-marked comparator for all injections areas except the chin.
Melanie Palm et al. 2021 [45]	PLLA and sterile water	RCT	80	59/21	M—4 F—76	Nasolabial fold	Hypodermis, 25G needle	AE: Seven subjects (11.9%) in the study group and seven (33.3%) in the control group. The total AEs were headache (two in the control group and one in the study group), running nose (two in the control group), and perioral hypoesthesia (one each in the study group and in the control group). The longest duration was 166 days for papule in the control group and 11 days for herpes in the study group. One subject in the study group developed a papule, and one subject a node.	All associated AEs, most of which were of mild intensity, resolved within 1 week. After treatment for both groups, no action was required.	The test substance PLLA showed a similar effect as the control group in reducing NLF. Safety was not compromised with higher volumes of solution including lidocaine.
Ge Feng et al. 2023 [46]	HA	RCT	40	20/20	M—0 F—40	Nasolabial fold	Hypodermis, 27G 1/2-inch needle	AE (conventional and exploratory): swelling, redness, numbness, pain, discoloration, stiffness, hyperpigmentation, foreign body sensation. Specific numbers, percentages not to submit.	All AEs symptoms resolved within two weeks.	The efficacy and safety of the investigational method for NLF correction is similar to the traditional method. The method is superior to the traditional method by improving the midface region and reducing AE.
Hongyi Zhao et al. 2023 [47]	PCL and sodium hyaluronate gel	RCT	160	80/80	M—5 F—155	Nasolabial fold	Dermis, 27G needle	AE: 18—swelling test. group, 18—swelling and 18—color change in control group.	All AEs resolved and no sequelae were observed.	Injection of the test substance is effective and safe with moderate correction severity and severe NLF.
Sun Young Choi et al. 2020 [48]	HA, 2 products	RCT	62	62/62	M—0 F—62	Nasolabial fold	Linear injection technique —not specified how deep, needle (size not specified)	ISR (study group): 44—pain, 39—sensitivity, 31—swelling, 15—redness, 11—hemorrhage, 18—pigmentation. ISR (control group): 49—pain, 44—sensitivity, 35—swelling, 18—redness, 15—hematoma, 23—itching, 14—papule, 5—pigmentation.	All local reactions at the injection site were mild and short-lived resolved within 2 weeks without any intervention or treatment.	HA filler containing lidocaine can reduce pain in patients during NLF correction; the addition of lidocaine does not affect the efficacy and safety of HA filler.
Yun Xie et al. 2022 [49]	HA, 2 Products	RCT	175	175/175	M—9 F—166	Nasolabial fold	Linear injection tech.—not specified how deep, 27G 1/2 in. or 30G 1/2 inches needle	One subject (0.6%) experienced one associated AE (paresthesia, control med. injection) that began 89 days after. The most commonly observed symptoms were tenderness, swelling, and pain.	The average duration of ISR was 2–5 days. Paresthesia disappeared within 8 days. without intervention.	The test material was not inferior to the control material. The study results show that the study material is safe and effective for up to 12 months.
Sai Luo et al. 2020 [51]	Extracellular matrix/stromal vascular fraction gel—processed in-house fat	NRT	33	-	M—8 F—25	Area under eyes	Suborbicuaris oculi fat, 21G thick needle	Less obvious fibrotic nodules appeared in 14 (21.2%) sides in 20 days; 28 patients (84.8%) had mild swelling 2 weeks on the run and three (9.1%) had petechiae at the injection site.	Fibrous nodules disappeared within 3–6 months; petechiae resolved within 1 week.	The high retention rate of the test material indicates that it can be an effective solution for corrections of the suborbicuar region. Further studies are needed to determine long-term efficacy and safety.
Sapna Vadera et al.2021 [52]	Fillers with lower and higher G primes	RCT	30	15/15	-	Lower eyelid	Dermis, lateral injection technique, 30G needle	AE (in the research group): 1—hemorrhage; AE (in the comparison group): 4—hemorrhage, 1—hematoma, 2—tyndall effect (color changes).	Not available	Compared to the standard medial approach, the lateral approach has been shown to be more effective. There were fewer bruises and major complications risks and incidence using the lateral approach. More research needed.

AB—Antibiotics, HA—Hyaluronic Acid, PCL—Polycaprolactone, HA_RK_—filler containing 20 mg/mL Hyaluronic Acid, HA_JUS_—filler containing 24 mg/mL Hyaluronic Acid, SP-HAL (Restylane Silk)—Small-Particle Hyaluronic Acid Filler with 0.3% Lidocaine, CPM-HA (Belotero Balance)—Cohesive Polydensified Matrix Hyaluronic Acid Filler, ISD—Ial System Duo, SF-01 (SYB filler*^®^*, Samyang Holdings Co., South Korea)—PCL-Based Fillers, Ellansé-M*^®^* (Sinclair Pharma, Ltd., London, UK)—PCL-based filler, VYC-25L (Juvéderm Volux, Allergan plc)—Hyaluronic Acid Soft-Tissue Filler with Lidocaine, BB—Belotero Basic/Balance, VYC-12 (Juvéderm*^®^* Volite™, Allergan plc)—Cross-Linked Hyaluronic Acid Injectable Gel, VYC-15L (Juvéderm Volbella XC; Allergan Aesthetics, an AbbVie Company, Irvine, CA, USA)—15 mg/mL Hyaluronic Acid Filler with 0.3% *w*/*w* Lidocaine, YC-12L (Allergan Aesthetics, an AbbVie company, Irvine, CA, USA)—Hyaluronic Acid filler that contains 12 mg/mL Hyaluronic Acid with 0.3% *w*/*w* lidocaine, DACS—Digital Analysis of the Cutaneous Surface, BA—Botulin A, RPC—Rapidly Polymerizing Collagen, VYC-20L—Juvederm Voluma with Lidocaine, SHAPE-NVL—Neuramis Volume Lidocaine, CE (Europian Comunity) marking—indicates that a product has been assessed by the manufacturer and deemed to meet EU safety, health and environmental protection requirements, PLLA—Poly-L-Lactic Acid, DLMR01 (DexLevo; Inc.)—PCL-based Dermal Filler, RCT—Randomized Controlled Trial, NRT—Non-randomized Trial, NLF—Nasolabial Fold, AE—Adverse Event, ISR—Injection Site Reaction, F—Female, M—Male.

## Data Availability

Data described in the manuscript, code book, and analytic code will be made available upon request pending application and approval from the corresponding author.

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
