# Peer review of "Safety and Potential Complications of Facial Wrinkle Correction with Dermal Fillers: A Systematic Literature Review"

_medicina, 2024, doi:10.3390/medicina61010025_

Round 1

Reviewer 1 Report

Comments and Suggestions for Authors

Thank you for the opportunity to review the manuscript entitled “Safety And Potential Complications of Facial Wrinkle 2 Correction with Dermal Fillers: A Scientific Systematic Review”. It discusses an important and very relevant subject.

While the scientific methods seem appropriate, my main concern is that the manuscript needs thorough editing.

Abstract:

“ In thirty-eight articles, 24 which involved 3967 participants, a total of 8795 complications were reported”. How can the number of complications be larger than the number of participants, and yet the authors conclude that “Facial 32 wrinkle correction procedures are generally safe...”?

The text needs editing. For example: The logic behind the following sentence is not clear: “ Complications from using dermal fillers are not common, based  on the annually increasing number of procedures in the United States” (LINES 45-46).

Explain abbreviations at the 1st time they are mentioned: i.e., PICCO

Introduction:

The description of each material by the same categories (Lines 58-155) is both too repetitive and not appropriate. This section can instead be put into a table and referred to in the introduction.

Results

The section :” Overview of Complications Reported in Studies” is very hard to follow, as it is composed of long lists of complications. Here, the complications need to be grouped into more broad groups, and the full details shown in table/s.

Discussion and conclusions are fine.

Minor issues:

The text needs editing. For example: The logic behind the following sentence is not clear:  “ Complications from using dermal fillers are not common, based  on the annually increasing number of procedures in the United States” (LINES 45-46).

Explain abbreviations at the 1st time they are mentioned: i.e., PICCO

Author Response

Comment no.1

Answer: In this study, we included only little complications, such as swelling, in our evaluation of dermal fillers. For instance, swelling and minor hematomas are commonly observed following injections. As a result, a single patient may experience multiple occurrences of these minor complications. Consequently, the reported complication rates can exceed the total number of patients included in the study. This distinction is important to accurately interpret the data presented.

Comment no.2

Answer: We made the statement more clear.

Comment no.3

Answer: The abbreviation PICO is clarified in the following text, and subsequent abbreviations are explained as they are introduced throughout the manuscript for clarity and consistency.

Comment no.4, 5

Due to the presence of an extensive table already included in our text, we were unable to incorporate additional tables. In the introduction, the classification of complications is presented as it would typically appear in a textbook. Grouping early complications alongside late complications would not resolve the issue of length; instead, it would result in similarly extensive content, only formatted within a bordered table.

Reviewer 2 Report

Comments and Suggestions for Authors

The article is well-structured and thoughtfully developed, with clear organization and detailed content throughout. To further enhance the paper, I would suggest expanding the introduction to provide a more comprehensive overview. Specifically, an emphasis on the epidemiological use of fillers could add valuable context, allowing readers to better understand the prevalence, demographic trends, and usage patterns associated with filler treatments in various populations. This expanded introduction would set a stronger foundation for the discussion and highlight the broader significance of the topic within the field.

Author Response

Dear Reviewer,

Thank you for your valuable comments and feedback. We appreciate your suggestions and would like to explain our decision regarding the introduction section. We aim to maintain a concise introduction to ensure that the reader remains focused and does not become overwhelmed by an excessive amount of information. Our intention is to provide a clear and engaging entry point to the paper, facilitating an effective understanding of the key points without unnecessary complexity.

We trust this approach will enhance the overall readability and impact of the manuscript.

Kind regards,

authors

Reviewer 3 Report

Comments and Suggestions for Authors

Line 47/48 Please clarify whether you mean 16.0% etc of the complications or of the overall cases.

Table 4: Please explain the abbreviations below the table.

On the whole the study is well written and interesting. I recommend acceptance taking into account the suggestions mentioned above.

Author Response

Comment 1: Line 47/48 Please clarify whether you mean 16.0% etc of the complications or of the overall cases. 

Answer to the comment: The complications outlined above represent the most frequently encountered adverse effects associated with the use of dermal fillers.

Comment 2: Table 4: Please explain the abbreviations below the table.

Answer: Thank you for the comment, it will be done.

Reviewer 4 Report

Comments and Suggestions for Authors

The manuscript presents a well-structured study. 

However, there is a minor inconsistency in the data presentation. In lines 261 and 262, the total number of patients (3967) does not match the sum of patients divided by gender (243 men and 3621 women, totaling 3864). 

With this minor revision addressed, the article is highly suitable for publication.

Author Response

Dear Reviewer,

Thank you for taking the time to review our manuscript and for providing your valuable feedback. We have carefully considered your comments and made the necessary corrections to address the issue you identified. Specifically, we have revised the text on line 261 to ensure greater accuracy and clarity.

We sincerely appreciate your thoughtful input, which has contributed to improving the quality of our work.

Kind regards,

authors